# Effect of Controlling Light on Cashmere Growth and Harmful Gas Parameters in Shanbei White Cashmere Goats

**DOI:** 10.3390/ani13060995

**Published:** 2023-03-09

**Authors:** Wenyuan Cui, Changlong Lin, Yuyang Liu, Zhixin Qiu, Wenrui Gao, Chunxin Wang, Yulin Chen, Yuxin Yang

**Affiliations:** 1Animal Science and Technology, Northwest A&F University, Xianyang 712100, China; 2Hengshan District Animal Husbandry Bureau, Yulin 719000, China; 3Jilin Academy of Agriculture Sciences, Gongzhuling 136100, China

**Keywords:** goat, cashmere, light control, photoperiod, harmful gas

## Abstract

**Simple Summary:**

Existing studies have paid attention to the effect of light-controlled conditions on the increase in cashmere production but changes in the environmental parameters of goat houses have been ignored. In this study, we monitored concentrations of CO2 and NH3 in a light-controlled goat house while evaluating the effect of light-controlled conditions on the increase in cashmere production. The results show that the concentrations of harmful gases under short photoperiod treatment are harmful to the health of goats. This study will provide a reference for the formulation of ventilation schemes in light-controlled goat houses.

**Abstract:**

The quality and yield of cashmere closely affect the economic benefits of cashmere goat farming. Studies have shown that controlling light can have an important impact on cashmere but can also affect the concentration of harmful gases. In order to explore the impact of a short photoperiod on the growth of cashmere and harmful gases in goat houses, 130 female (non-pregnant) Shanbei white cashmere goats, aged 4–5 years with similar body weights, were randomly divided into a control group and a treatment group, with 65 goats in each group. The dietary nutrition levels of the experimental goats were the same, and completely natural light was used in the control group; the light control group received light for 7 h every day (9:30–16:30), and the rest of the time (16:30–9:30 the next day) they did not receive light. The light control treatment was carried out in a control house, and the gas content was analyzed. It was found that a shortened period of light exposure could increase the annual average cashmere production by 34.5%. The content of each gas has a certain functional relationship with the measurement time period, but at the same time, we found that the content of NH3 also changes seasonally. In summary, the use of shortened light periods when raising cashmere goats can significantly increase cashmere production and quality, but at the same time, it will increase the concentration of harmful gases in the goat barn, and ventilation should be increased to ensure the health of the goats and the air quality in the barn.

## 1. Introduction

Cashmere products are mostly dispersed worldwide and are well known for their fineness, warmth, softness and elegance [1] China is the largest cashmere producer in the world [2]. The Shanbei white cashmere goat, a double-coated (cashmere and mutton) species famous for its high production of luxurious fiber, is an excellent cashmere goat breed bred in China [3].

Cashmere growth is seasonal, which arises from circannual changes in natural photoperiods [4]. The cashmere of Shanbei white cashmere goats grows from July to February and sheds in May [3]. Short photoperiods can affect the morphological development of hair follicles, and the structural characteristics of hair follicles directly affect the yield and quality of cashmere [5]. In the process of breeding cashmere goats, the cashmere yield and quality were previously improved by shortening the illumination time to 7 h [5].

Although the technology of controlling light has the advantages of high efficiency, low cost and strong operability, it is not largely popularized or applied at present, especially when considering the control of harmful gas content in goat houses. The concentrations of harmful gases such as carbon dioxide (CO_2_) and ammonia (NH_3_) will increase in a light-controlled goat house due to the long-term enclosed nature of this type of house. This may make goats breathless, more likely to suffer from respiratory diseases, damage their health and reduce their production performance [6,7]. Existing studies have paid attention to the effect of light-controlled conditions on the increase in cashmere production but have ignored the changes in the environmental parameters of goat houses. In this study, we monitored concentrations of CO_2_ and NH_3_ in a light-controlled goat house while evaluating the effect of light-controlled conditions on increases in cashmere production to promote the popularization of this technology, and we also provided a reference for the formulation of ventilation schemes in light-controlled goat houses.

## 2. Materials and Methods

### 2.1. Experimental SITE and Animals

The experiment was performed in a Shanbei white cashmere goat breeding farm in Hengshan District, Shaanxi Province, China (latitude 37°38′ N, longitude 109°12′ E, altitude 1487 m). The goat houses were brick wall and steel frame structures. Detailed physiological conditions of goats are shown in Table 1: One hundred and thirty female (non-pregnant) Shanbei white cashmere goats, aged 4–5 years old with similar weights and body conditions, were randomly allocated into a control group or a treatment group, with sixty-five goats per group. The weights of all the goats were measured before the start of the experiment, and it was found that the weight difference between the control group and treatment group was not significant (*p* > 0.05).

### 2.2. Weather in Experiment

The relevant weather parameters in the experimental locations are expressed in Table 2. These included the mean monthly rainfall and the mean monthly temperature variation.

### 2.3. Feeding and Management

The diets of the goats in the control group and the treatment group were the same. The ratio of concentrate to roughage was 4:6, and the goats were fed regularly and quantitatively twice a day with free drinking. During 9:30–16:30, the goat houses in the control group and the treatment group maintained natural ventilation. Efforts were made to prevent diseases within the goats and disinfect the goat houses before the experiment started.

### 2.4. Experimental Design

The goats underwent normal shearing at the beginning of May in 2015. In the light control group (treatment group), goats were put in goat house from 16:30 to 9:30 a.m. the next day to avoid light from 15 May to 15 October 2015. The light intensity in the house was less than 0.1 Lx. There was a 7-day adaptation period before the start of the formal experiment, and the illumination time gradually decreased to the set time. The control group was free to move all day. From 16 May to 22 September 2015, the main harmful gas (CO_2_ and NH_3_) parameters in the goat house were recorded dynamically using a multi-channel gas analyzer. Three gas monitoring points were set up in the goat houses in the control group and the light control group. The monitoring points were evenly distributed on the central axis of the goat house at the same interval, 1.5 m above the ground. Additionally, there was an outdoor monitoring point. Switches in which monitoring point the data were recorded at occurred at 10-min intervals; this cycle was completed every 70 min.

### 2.5. Sample Collection and Indicator Determination

Immediately after the end of the light control experiment, the cashmere samples were collected. Five goats with similar nutritional statuses were each selected from the treatment group and the control group. The cashmere and wool were collected from the thigh, shoulder, back, abdomen and side of each goat, and then, all the cashmere and wool from the ten goats were collected to measure the weight. The second sampling procedure took place at the end of April in 2016. Ten goats were sampled again. Here, five goats with similar nutritional statuses in the treatment group and the control group were each selected to collect cashmere and wool from the same five areas, and all the cashmere and wool from the ten goats were collected to measure the weight. After washing and drying the cashmere and wool taken from the five areas, the cashmere was separated manually, and its weight was determined. The total cashmere yield of each goat was calculated according to the ratio of cashmere and total cashmere and wool weight. The straightened length of the cashmere was measured. The concentrations of the main harmful gases (CO_2_ (%) and NH_3_ (ppm)) in the goat house were monitored using the multi-channel gas analyzer. The specific methods are described below.

Cashmere and wool measurement method: After the test, a shearer was used to cut all the wool of the goat.

Cashmere and wool weight: Selected goats with similar body conditions in the test and control flocks were clipped, and the initial weight of the coarse hair mixed with the vellus hair samples was determined using an analytical balance.

Stretched length of cashmere fiber: The measurements were performed according to the measurement method given in DB65/T4167-2018 standard using a portable 24/7 velum length rapid detection unit manufactured by Tiancheng Zhongding Technology Co., Ltd. (Tianjin, China).

Separation of cashmere and wool: Manual isolation was used in this study.

### 2.6. Statistical Analysis

The whole-stage data were divided into two periods: the light period (9:30–16:30) and dark period (16:30–9:30 the next day). Data were first classified and analyzed using Excel 2013. SPSS 20.0 was used for the *t* tests. The results were expressed as the mean and SD. A *p* < 0.05 indicated a significant difference. Line charts were drawn to reflect changes in the trends of the gas parameters.

## 3. Results

### 3.1. Effects of Short Photoperiods on Cashmere Growth and Quality in Shanbei White Cashmere Goats

The effect of short photoperiods on the cashmere yield and the straightened lengths of cashmere fibers of Shanbei white cashmere goats is shown in Table 3. The analysis results from the cashmere samples collected in October show that compared with the control group, the mixed weight of cashmere and wool, the yield of cashmere and the straightened lengths of cashmere fibers in the treatment group increased significantly (*p* < 0.05), by 23.1%, 57.3% and 23.8%, respectively. The analysis results of cashmere samples collected in April of the next year show that compared with the control group, the mixed weight of cashmere and wool, the yield of cashmere and the straightened lengths of cashmere fibers in the treatment group increased significantly (*p* < 0.05), by 13.2%, 18.8% and 7.7%, respectively. In addition, compared with the control group, after treatment, the ratio of cashmere to wool clearly improved (*p* > 0.05). We found that short photoperiods significantly increased the annual average cashmere yield by 34.5% (*p* < 0.05).

### 3.2. Variation Trend in CO_2_ Concentration in the Goat House

The average value of gas concentration was calculated by day, and the trend in CO_2_ concentration during the whole test period was determined (Figure 1). The average CO_2_ concentration in the goat house in the treatment group was significantly higher than that in the control group and the outdoor control point throughout the day. The range of CO_2_ concentration in the outdoor air was 0.02–0.03%, that in the treatment group was 0.07–0.16% and that in the control group was 0.02–0.04% (Figure 1a). During the light period, the average CO_2_ concentrations in the goat house in the treatment group and control group were higher than that at the outdoor control point. The range of CO_2_ concentration in the outdoor air was 0.02–0.03%, that in the treatment group was 0.03–0.05% and that in the control group was 0.02–0.04% (Figure 1b). During the dark period, the average CO_2_ concentration in the goat house in the treatment group was significantly higher than that in the control group and at the outdoor control point. The range of CO_2_ concentration in the outdoor air was 0.02–0.03%, that in the treatment group was 0.09–0.29% and that in the control group was 0.03–0.05% (Figure 1c).

In this study, the dates from 5.21 to 6.20 comprised the first month, June, the dates from 6.21 to 7.20 comprised the second month, July, the dates from 7.21 to 8.20 comprised the third month, August, and the dates from 8.21 to 9.20 comprised the fourth month, September. According to the time sequence of a day, an hour was taken as the time period, and after obtaining an average of all the data in each time period, Figure 2a was created. There was no clear trend in CO_2_ concentration variation between the different months. It was observed that the concentration of CO_2_ decreased rapidly after the door of the goat house was opened at 9:30, until it became stable. After the door of the goat house was closed at 16:30, the concentration of CO_2_ increased slowly within 3 h and gradually stabilized. The variation in the CO_2_ concentration in each individual month followed this rule. The data regarding the light control group were organized according to the time sequence from 0:00 to 23:59, and Figure 2b was created.

According to the variation in the CO_2_ concentration obtained from this analysis, the gas data from the light-controlled goat house from 9:30 to 11:00 and 16:30 to 19:30 were extracted. The results show that the change in the CO_2_ concentration in the goat house in the treatment group after the light period began at 9:30 accords with the logarithmic function: *Y* = −0.033 ln(*X*) − 0.0651 (*X* indicates the time after opening the door; *Y* indicates the concentration of CO_2_; Figure 3a). The change in the CO_2_ concentration after 16:30 accords with the power function relationship: *Y* = 0.4368*X*^0.3834^ (*X* indicates the time after closing the door; *Y* indicates the concentration of CO_2_; Figure 3b).

### 3.3. Variation Trend in NH_3_ Concentration in Goat House

The average value of gas concentration was calculated by day, and the trend in NH_3_ concentration during the whole test period was determined (Figure 4). The average NH_3_ concentration in the goat house in the treatment group was significantly higher than that in the control group and at the outdoor control point throughout the day. The NH_3_ concentration in the outdoor air ranged from 3.3 to 9.0 ppm, that in the treatment group ranged from 7.1 to 15.9 ppm and that in the control group ranged from 2.5 to 8.7 ppm (Figure 4a). During the light period, the average NH_3_ concentration in the goat house in the treatment group was higher than that at the outdoor control point and in the control group. The NH_3_ concentration in the outdoor air ranged from 0.6 to 7.8 ppm, that in the treatment group ranged from 3.5 to 10.2 ppm and that in the control group ranged from 1.0 to 7.5 ppm (Figure 4b). During the dark period, the average NH_3_ concentration in the goat house in the treatment group was significantly higher than that in the control group and at the outdoor control point. The NH_3_ concentration in the outdoor air ranged from 3.8 to 10.3 ppm, that in the treatment group ranged from 8.1 to 20.3 ppm and that in the control group ranged from 3.6 to 10.2 ppm (Figure 4c).

According to the time sequence of a day and by taking an hour as a time period, we obtained an average of all the data in each time period; Figure 5a was created from this. During the dark period, the NH_3_ concentration decreased from June to September, but during the light period, it was lowest in August. It was observed that the concentration of NH_3_ decreased rapidly after the door of the goat house was opened at 9:30, until it became stable. After the door of the goat house was closed at 16:30, the concentration of NH_3_ increased slowly within 3 h and gradually stabilized. The variation in the NH_3_ concentration in each individual month adhered to this rule. The data from the light control group were organized according to the time sequence from 0:00 to 23:59, and Figure 5b was created.

## 4. Discussion

### 4.1. Effects of Short Photoperiods on Cashmere Growth and Quality in Shanbei White Cashmere Goats

The yield and length of cashmere are important economic traits, which are influenced by varieties, ages, nutritional level and environment [8]. Regarding environmental factors, in as early as 1987, Australian scholars found that wild goats experience seasonal changes to their wool [9]. Changes in the time goats are exposed to light affect the growth rate, length and diameter of cashmere. Studies have shown that the influence of light on cashmere growth is related to animals’ biological clocks. Changes to photoperiods affect hormone secretion in animals and further regulate the periodic growth and development of hair follicles [4]. Melatonin secretion is one of the main factors. The pineal gland inhibits melatonin secretion in the daytime and synthesizes melatonin in large quantities at night; melatonin directly influences cashmere growth [10]. Short photoperiods were shown to increase melatonin secretion and promote cashmere growth. In this study, in the light control group, a significant increase in the lengths of cashmere fibers and the production of cashmere was observed, which was consistent with previous studies carried out on goats [2,5].

### 4.2. Effects of Light Control on the Gas Environment of the Goat House

The main harmful gases in ruminant houses include CO_2_ and NH_3_, which are produced via animal respiration, the decomposition of organic matter in feces and urine and the fermentation of gastrointestinal microorganisms. CO_2_ itself is non-toxic; it mainly causes harm by causing hypoxia in livestock. Animals in a long-term hypoxic environment can experience mental depression, the loss of appetite and reduced productivity. The concentration of CO_2_ is often used as an important index to evaluate the air quality of livestock houses [11] NH_3_ can bind to hemoglobin and reduce the oxygen-carrying capacity of hemoglobin, which leads to anemia and low resistance in animals. A high concentration of NH_3_ can even cause damage to the central nervous system [6].

The concentration of harmful gases in goat houses is affected by animal feeding density, temperature and ventilation. In this study, it was observed that the concentrations of CO_2_ and NH_3_ during dark periods were higher than those during light periods in the experiment group, and the concentrations of gas in the experiment group were significantly higher than those in the control group and at the outdoor monitoring point. This is because during dark periods, goats are in a closed environment for a long time, the density of goats is high, the temperature increases and the lack of appropriate ventilation equipment leads to an increase in harmful gas concentration. In the meantime, the movement of goats causes dust and promotes NH_3_ volatilization from feces and urine fermentation into the air [12]. Due to the ventilation and free movement of goats, here we found that the trend regarding the change in gas concentration in the control group was consistent with that at the outdoor monitoring point.

At present, there is no air quality standard for goat houses in China. According to the Environmental Quality Standards for Livestock and Poultry Farms of China, the concentration of CO_2_ should not exceed 1500 mg/m^3^, and the concentration of NH_3_ should not exceed 20 mg/m^3^ according to the air quality of cowsheds. In this study, the average concentration of CO_2_ in the light-controlled goat house was 3734 mg/m^3^ (0.1901%) during the dark period, far exceeding the standard limit concentration. The NH_3_ concentration did not exceed the standard of cowsheds. However, goats are smaller than cattle, and whether NH_3_ is harmful to goats under light control requires further study. In conclusion, light control accelerates the accumulation of harmful gases in light-controlled goat houses. In order to ensure the health of goats and reduce the stress reactions they experience, the concentration of harmful gases should be reduced by increasing ventilation or by reasonably reducing the feeding density.

## 5. Conclusions

In the cashmere non-growing period for Shanbei white cashmere goats, the cashmere length and the annual cashmere yield significantly increased when the light duration period was shortened, while the concentrations of CO_2_ and NH_3_ in the goat house increased. The concentration of CO_2_ presents a functional relationship in the period of light control. Combined with the changing laws regarding gas parameters in goat houses, it is important to increase the ventilation volume and reduce the harmful gas content during high-temperature seasons and light control periods to ensure the health of goats. At the same time, it is necessary to establish a more accurate gas change model in goat houses, using light control to guide production in future research.

## Figures and Tables

**Figure 1 animals-13-00995-f001:**
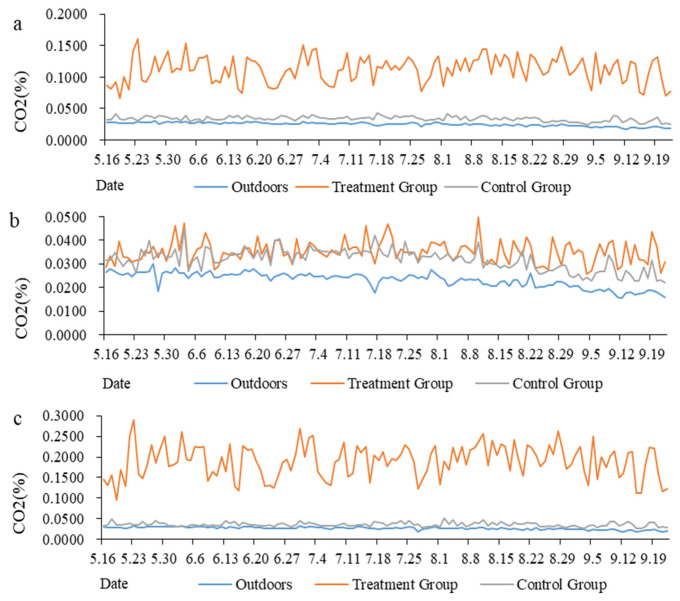
The variation trends in CO_2_ concentration throughout the experimental period: (**a**) The variation trends in CO_2_ concentration all day; (**b**) The variation trends in CO_2_ concentration during the light period (9:30–16:30); (**c**) The variation trends in CO_2_ concentration during the dark period (16:30–9:30 next day).

**Figure 2 animals-13-00995-f002:**
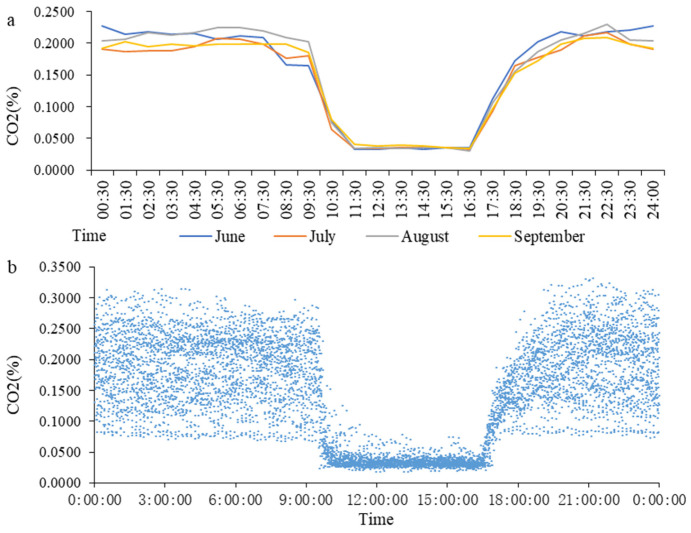
The variation trends in CO_2_ concentration in the treatment group. (**a**): The variation trends in CO_2_ concentration in different months; (**b**): summary of data of CO_2_ concentration.

**Figure 3 animals-13-00995-f003:**
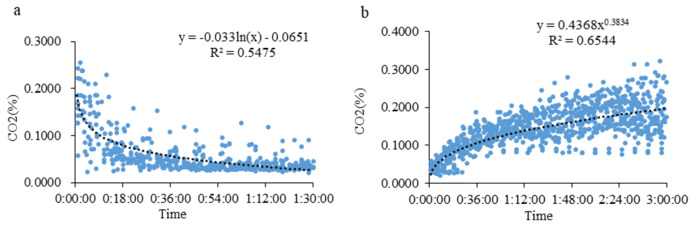
Mathematical model of CO_2_ concentration change in the treatment group after the door was opened or closed. (**a**) 9:30–11:00 light-controlled goat house CO_2_ concentration change model; (**b**) 16:30–19:30 light-controlled goat house CO_2_ concentration change model.

**Figure 4 animals-13-00995-f004:**
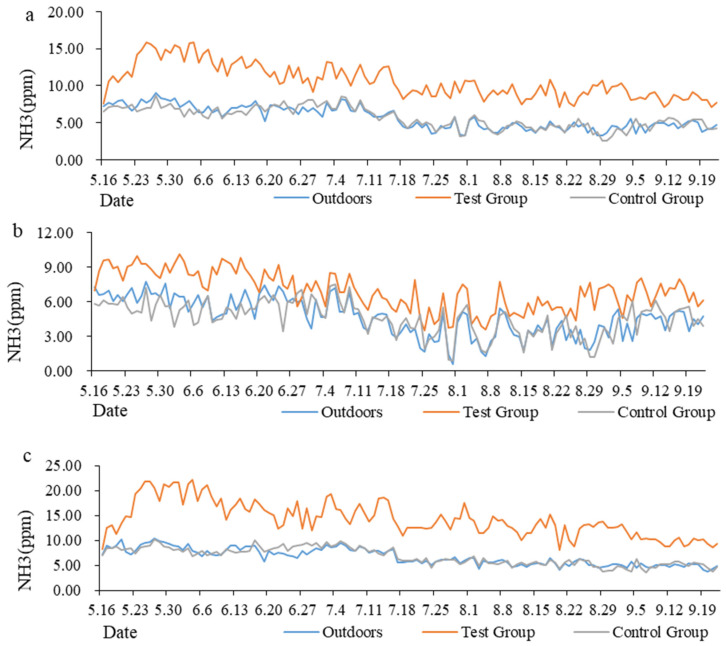
The variation trends in NH_3_ concentration throughout the experimental period. (**a**) The variation trends in NH_3_ concentration all day; (**b**) The variation trends in NH_3_ concentration during the light period (9:30–16:30); (**c**) The variation trends in NH_3_ concentration during the dark period (16:30–9:30 next day).

**Figure 5 animals-13-00995-f005:**
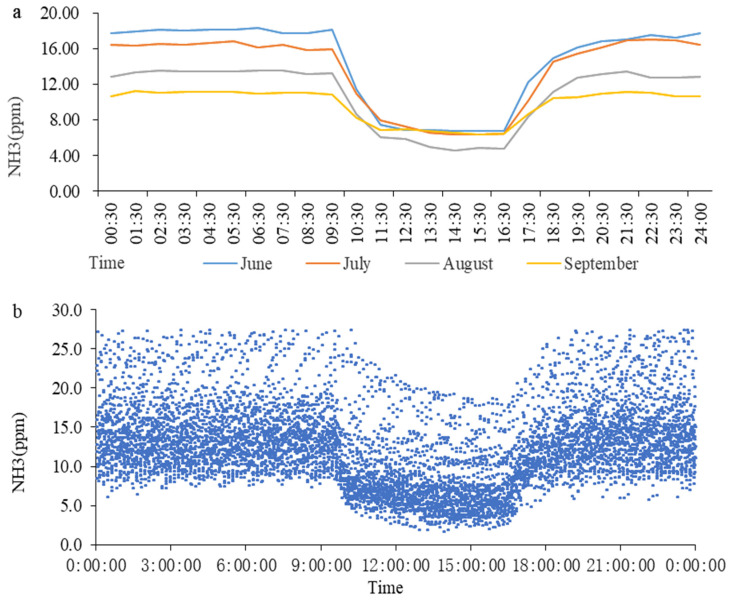
The variation trends in NH_3_ concentrations in the treatment group. (**a**) The variation trends in NH_3_ concentration in different months; (**b**) Summary of data regarding NH_3_ concentration.

**Table 1 animals-13-00995-t001:** Physiological statuses of goats selected for the experiment.

Item	Control	Treatment	Average SD	*p* Values
Animals (n)	65	65	-	-
Period	Non-pregnancy	Non-pregnancy	-	-
Age (Year)	4–5	4–5	-	-
ABW(kg)	47.36	46.08	3.32	0.87

Note: - indicates not applicable.

**Table 2 animals-13-00995-t002:** The mean monthly temperature and precipitation at experimental sites.

Month	Average Temperature		Average Rainfall (mm)
	Minimum Temperature (°C)	Maximum Temperature (°C)	
June	18.2 ± 2.4	31.8 ± 3.0	12.6
July	18.0 ± 2.5	32.0 ± 2.8	72.1
August	17.3 ± 1.9	28.7 ± 2.5	200.8
September	12.1 ± 2.7	24.0 ± 2.2	80.3
October	3.1 ± 2.8	16.5 ± 1.9	34.0

**Table 3 animals-13-00995-t003:** The effect of short photoperiods on cashmere growth and quality.

Item	October				April of the Next Year			
	Treatment	Control	Average SD	*p* Values	Treatment	Control	Average SD	*p* Values
Cashmere and wool weight (g)	723.0	587.5	98.8	0.030	1252.4	1106.9	133.8	0.021
Cashmere weight (g)	302.7	192.4	80.3	0.049	589.7	496.4	72.6	0.024
Fluff ratio (cashmere:wool)	1.44	1.10	0.38	0.114	0.82	0.62	0.51	0.132
Stretched length of cashmere fiber (cm)	6.20	5.01	0.77	0.023	9.80	9.10	1.4	0.044
Wool weight (g)	420.3	307.7	18.3	-	662.7	610.5	20.9	-
Cashmere weight of the year (g)	-	-	-	-	755.1	561.5	80.1	0.007

## Data Availability

The raw/processed data required to reproduce these findings cannot be shared at this time as the data also form part of an ongoing study.

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
