# Peer review of "Effect of Controlling Light on Cashmere Growth and Harmful Gas Parameters in Shanbei White Cashmere Goats"

_animals, 2023, doi:10.3390/ani13060995_

Round 1
Reviewer 1 Report
1.Why should cashmere and wool be weighed together? As we all know, cashmere of cashmere goat is a scarce resource, so cashmere and wool should be weighed separately to compare the yield before and after light respectively.
2.The control group and the treatment group were represented by differences in light duration. In the result and discussion section, the enclosed space of the short light group is also described. So, what are the variables between the two experiments? It's not just the length of light that's different in the enclosed space.
3.In the sample selection, randomly in each group only picked out the five goat, whether there is a sample size too little can't enough, not fully support the conclusion of article
4.The article does not explain how to measure the strength of cashmere, which instruments are used, etc.
Author Response
Dear Reviewer 1: Thanks for your valuable suggestions on my manuscript, which have greatly improved the quality of Comments and Suggestions for Authors. 1.Why should cashmere and wool be weighed together? As we all know, cashmere of cashmere goat is a scarce resource, so cashmere and wool should be weighed separately to compare the yield before and after light respectively. RE: We have added wool data to the test results according to your suggestion. 2.The control group and the treatment group were represented by differences in light duration. In the result and discussion section, the enclosed space of the short light group is also described. So, what are the variables between the two experiments? It's not just the length of light that's different in the enclosed space. RE: In the experiment, it is divided into treatment group and control group. The treatment group goat is enclosed in the light control room from 16:30 to 9:30 the next day, and is given low light level, and the doors and windows are closed. At this time, compared with the control group, the air flow level is also significantly decreased, and the indoor temperature is also relatively higher, while the control group is located in the sheepfold for free feeding and free light all day. Therefore, the difference between the two is not only in light, temperature and air circulation, but also in water products, and the feeding density of goats is relatively larger at this time. 3.In the sample selection, randomly in each group only picked out the five goat, whether there is a sample size too little can't enough, not fully support the conclusion of article RE: We only selected 5 goats in each group (10 in total) because it was the middle of October at the end of the experiment. At this time, the weather at the experiment site was cold(The lowest temperature is close to 0 ℃), and the sheep who had removed the wool had shivering, shivering and other cold-resistant reactions. Therefore, we only selected 10 goats for animal welfare and comprehensive consideration of the experiment. 4.The article does not explain how to measure the strength of cashmere, which instruments are used, etc. RE: We have added detection methods to the article.Reviewer 2 Report
The paper deals with the effects of photoperiod on growth of cashmere and harmful gases in goat barns. Introduction is concisely and well written. The results of the obtained research were properly described.
Comments:
- line 117 - Is October listed correctly here? Shouldn't April be listed here?
- line 169 - Part of the sentence is missing. Please check.
- line 307 - Replace sheep with goats.
- Discussion - I recommend discussing the section “Effects of light control on CO2 and NH3 concentrations in goat house” with several authors.
Author Response
Dear Reviewer 2:
Thanks for your valuable suggestions on my manuscript, which have greatly improved the quality of Comments and Suggestions for Authors.
- line 117 - Is October listed correctly here? Shouldn't April be listed here?
RE: Thank you for your correction. We have corrected it.
- line 169 - Part of the sentence is missing. Please check.
RE: We found the problem and corrected it. Thank you for your suggestions.
- line 307 - Replace sheep with goats.
RE: We found the problem and corrected it. Thank you for your suggestions.
- Discussion - I recommend discussing the section “Effects of light control on CO2 and NH3 concentrations in goat house” with several authors.
RE: The author discussed your comments and optimized the content. Thank you for your suggestions.
Reviewer 3 Report
Dear authors,
Thank you for this piece of work which i read with great interest. I found it meritorious of being considered further to publication. However I feel it needs extensive amendment from your side, in my modest opinion.
The abstract should be amended as to the appropriate terms and sheep should be replaced by goats only.
The implication section should start directly from the second sentence. So please delete the first one.
The introduction is very good, short, well described and informative to the reader. However, please, check the citation style of the journal and proofread the english grammar.
In material and methods, some important data are missing. For instance average temperature across the period of observation. You correctly and precisely reported all information about geographical location, but that important datum is missing. Likewise rainfall (which later relates to harmful gaseous emissions...). Another very important datum, which is complementary to the usage and emission of NH3 and protein metabolism relates to the reproduction stage of goats. You enrolled non pregnant goats, aged 4-5 ys, and of similar weight (please, also amend in the text the expression to measure the weight and replace with weight assessment on a scale). Please, report the weight range. At this regards, I would suggest authors to adopt a schematic way for enrollment criteria of experimental animals dealing with weight, age, and reproduction phase. I believe that to justify the non-pregnant goat involvement in the trial, the reference to Cappai et al., 2019 Res Vet Sci, 123:84-90, doi: 10.1016/j.rvsc.2018.12.016. for the effect of Urea and protein metabolism and the consequent NH3 emission in the barn. I believe you selected non-pregnant goats also for this reason, right? Please, refer to this.
I would suggest to improve the table fashion, with easier reporting, like for instance averages without SD in the same cell and report instead the values of pooled-SD in a different column right before the p-value. In this way you would avoid to put the plus/minus symbol and too many numbers in a same cell, allowing a more readable content.
Discussion and conclusion are fine to me, with the request to improve english grammar and style.
Author Response
Dear Reviewer 3:
Thanks for your valuable suggestions on my manuscript, which have greatly improved the quality of Comments and Suggestions for Authors.
Thank you for this piece of work which i read with great interest. I found it meritorious of being considered further to publication. However I feel it needs extensive amendment from your side, in my modest opinion.
The abstract should be amended as to the appropriate terms and sheep should be replaced by goats only.
RE: We have corrected.
The implication section should start directly from the second sentence. So please delete the first one.
RE: We have checked the citation style of the journal and proofread the english grammar.
In material and methods, some important data are missing. For instance average temperature across the period of observation. You correctly and precisely reported all information about geographical location, but that important datum is missing. Likewise rainfall (which later relates to harmful gaseous emissions...). Another very important datum, which is complementary to the usage and emission of NH3 and protein metabolism relates to the reproduction stage of goats. You enrolled non pregnant goats, aged 4-5 ys, and of similar weight (please, also amend in the text the expression to measure the weight and replace with weight assessment on a scale). Please, report the weight range. At this regards, I would suggest authors to adopt a schematic way for enrollment criteria of experimental animals dealing with weight, age, and reproduction phase. I believe that to justify the non-pregnant goat involvement in the trial, the reference to Cappai et al., 2019 Res Vet Sci, 123:84-90, doi: 10.1016/j.rvsc.2018.12.016. for the effect of Urea and protein metabolism and the consequent NH3 emission in the barn. I believe you selected non-pregnant goats also for this reason, right? Please, refer to this.
RE: According to your suggestion, we have added the monthly average temperature and precipitation of the experimental site in the article, and also added the table of animal physiological conditions and the inclusion criteria with reference to your suggestions.
I would suggest to improve the table fashion, with easier reporting, like for instance averages without SD in the same cell and report instead the values of pooled-SD in a different column right before the p-value. In this way you would avoid to put the plus/minus symbol and too many numbers in a same cell, allowing a more readable content.
RE: We have modified the table according to your suggestion.